

# A new terrestrial snail species (Gastropoda: Bulimulidae) from the Región de Antofagasta, northern Chile

Juan Francisco Araya[1,2] and Abraham S.H. Breure[3,4]

[1] Programa de Doctorado en Sistemática y Biodiversidad, Departamento de Zoología, Facultad de Ciencias Naturales y Oceanográficas, Universidad de Concepción, Concepción, Chile
[2] Universidad de Atacama, Copiapó, Región de Atacama, Chile
[3] Royal Belgian Institute of Natural Sciences, Vautierstraat, Brussels, Belgium
[4] Naturalis Biodiversity Center, Leiden, The Netherlands

## ABSTRACT

A new species of *Scutalus* Albers, 1850 (Gastropoda: Bulimulidae), *Scutalus chango* sp. n., is described from a coastal area of northern Chile. Empty shells of this new species were found buried in sand and under boulders and rocks in the foothills of the Chilean Coastal Range at Paposo, Región de Antofagasta. This new species is distinguished from all other Chilean terrestrial snails by its slender shell with a flared and reflected aperture, and by the presence of a columellar fold. This is the first record of *Scutalus* in Chile, and the southernmost record for this endemic South American bulimulid genus. The presence of this species in Paposo highlights the need for further research and for conservation guidelines in coastal areas of northern Chile, which have comparatively high levels of biodiversity and endemism.

## INTRODUCTION

Terrestrial mollusks are one of the least well known invertebrate groups in Chile; knowledge of their diversity is based on comparatively few works, most of them from the 19th century, with a single, more comprehensive recent work (*Stuardo & Vega, 1985*), which lists 154 species in 14 families for all Chilean territories, including the Juan Fernández and Desventuradas Archipelagos as well as Easter Island. The Chilean terrestrial molluscs are mostly represented by species of the families Charopidae, Bulimulidae and Bothriembryontidae, most of them with very narrow distributions; these families show, in Chilean territories, high levels of endemism. Works which have reviewed terrestrial snails from the northern part of the country (characterized by its arid to hyper-arid landscapes) only include the studies done by *Philippi (1860)*, *Gigoux (1932)*, *Rehder (1945)*, *Biese (1960)*, *Breure (1978)*, *Valdovinos & Stuardo (1989)*, *Valdovinos & Stuardo (1988)*, *Craig (1992)*, *Miquel & Araya (2013)*, *Araya & Catalán (2014)*, *Araya (2015a)* and *Araya, Madrid & Breure (2016)*.

In the present study—part of ongoing work aimed at reviewing the terrestrial mollusks from northern and central Chile (*Araya & Aliaga, 2015*; *Araya, 2015b*; *Araya, 2016*)—we report a new terrestrial snail species, characterized by having a shell with an expanded

Corresponding author
Juan Francisco Araya,
jfaraya@u.uchile.cl, juan.araya@uda.cl

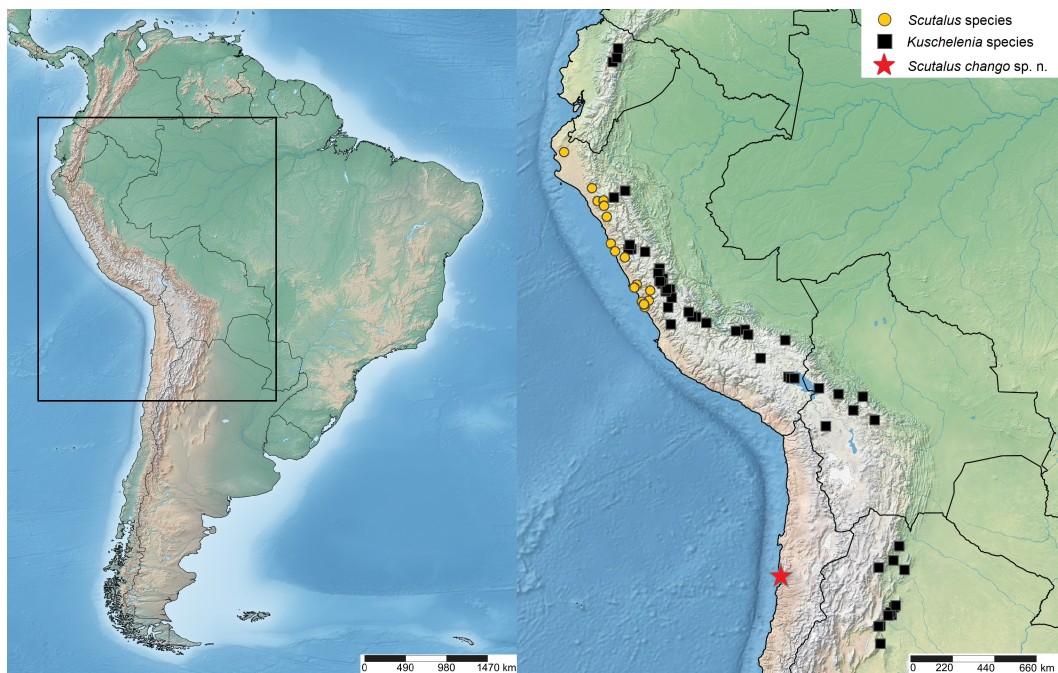

**Figure 1** **Location map.** Geographical location of *Scutalus chango* sp. n. (red star: type locality), Peruvian *Scutalus* species (yellow circles), and *Kuschelenia* species (black squares).

aperture and a columellar fold. It was collected buried in humus and sand among communities of arborescent cacti (*Eulychnia iquiquensis*), large succulent shrubs (*Euphorbia lactiflua*), and other xerophytic plants in a narrow area in the foothills of the Chilean Coastal Range north of Paposo, Región de Antofagasta, in northern Chile. This new species represents the southernmost record of the genus *Scutalus* Albers, 1850, a South American genus of the family Bulimulidae; this family was previously represented in Chile solely by the genus *Bostryx* Troschel, 1847 (*Valdovinos, 1999*).

## MATERIAL AND METHODS

### Material collection

Sixteen specimens, all of them empty shells, were collected buried in humus and under boulders and fallen rocks north of Paposo (24°55′S; 70°30′W, altitude 150–170 m; precise locality data available from first author on request), Región de Antofagasta, northern Chile (Figs. 1 and 2). The dimensions of the shells, measured with Vernier calipers ($\pm0.1$ mm) are depicted in Fig. 3; measurements are given in mm and include, when appropriate, the additional thickness of the lip. Type specimens are deposited in the collections of the Museo Paleontológico de Caldera (MPCCL), in Caldera, Chile and in the Santa Barbara Museum of Natural History (SBMNH) at Santa Barbara, USA. Field study permits were not required for this study and none of the species studied herein are currently under legal protection. The distribution map (Fig. 1) was prepared using SimpleMappr (*Shorthouse, 2010*).

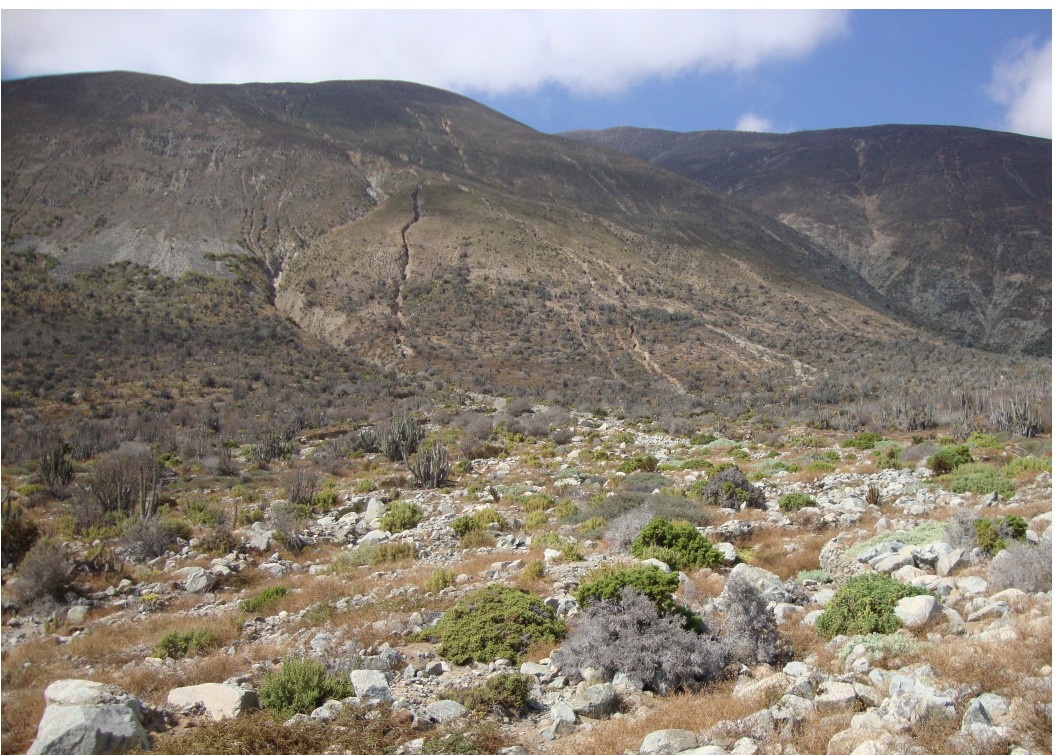

**Figure 2** Type locality and habitat of *Scutalus chango* sp. n.: under boulders at foothills of the Chilean Coastal Range (SE view), north of Paposo, Región de Antofagasta, northern Chile.

## Nomenclature

The electronic edition of this article conforms to the requirements of the amended International Code of Zoological Nomenclature, and hence the new name contained herein is available under that Code from the electronic edition of this article (*ICZN, 1999*; *ICZN, 2008*). This published work and the nomenclatural acts it contains have been registered in ZooBank, under the LSID urn:lsid:zoobank.org:pub:C9BE441E-6159-4973-888D-74660B2C25F3. The electronic edition of this work is available from the following digital repositories: PubMed Central, LOCKSS.

## RESULTS

### Systematic account

Superfamily Orthalicoidea Martens, 1860
Family Bulimulidae Tryon, 1867
Genus *Scutalus* Albers, 1850

**Diagnosis (Modified from *Breure, 1979*):** Shell elongate-ovate to rather globose or depressed conical; (broadly) perforate; solid. Whitish to brownish in color, often with darker spiral bands, with axial streaks or coalescent spots in some species. Surface granulate or with incrassate growth striae. Protoconch pit-reticulate. Whorls slightly convex. Aperture

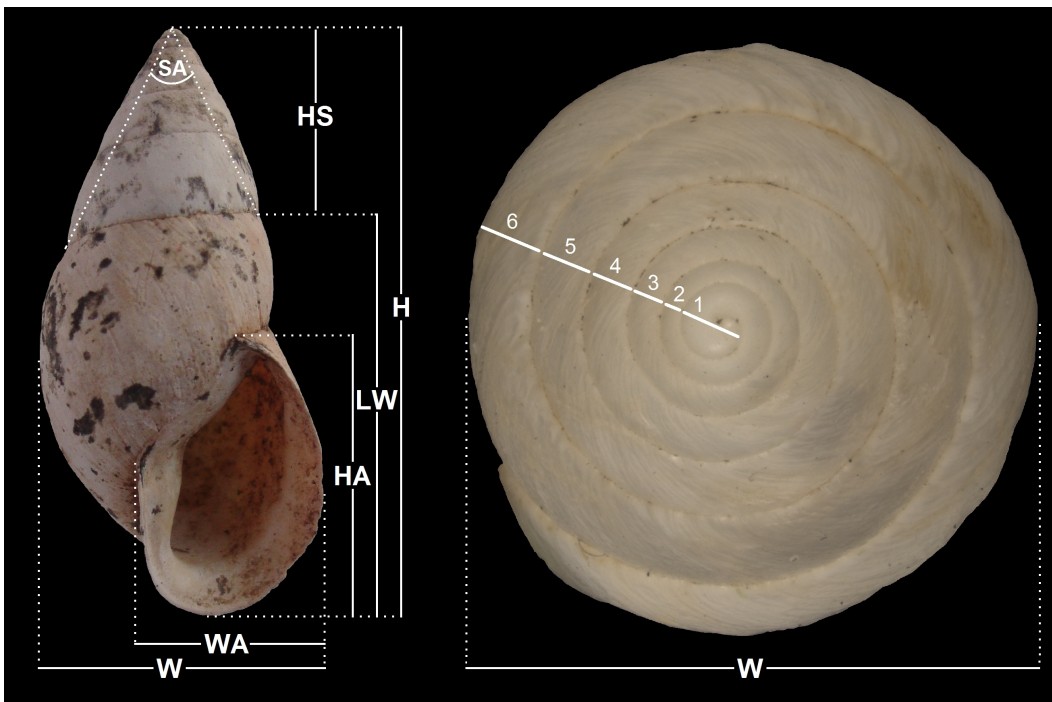

**Figure 3** **Measurements performed on shells.** Measurements taken on specimens and counting of whorls. Abbreviations are: diameter (D, maximum dimension perpendicular to H); height (H, maximum dimension parallel to axis of coiling); height of aperture (HA), height of last whorl (LW), height of aperture (HA), spire angle (SA), and width of aperture (WA).

(sub) ovate. Peristome more or less expanded. Columella in some species with a fold within the last whorl.

   **Type species** *Bulinus proteus* Broderip, 1832

   *Scutalus chango* new species (Figs. 4A–4Q, Figs. 5A–5D)

   **Diagnosis:** A species with a medium sized thick and elongated shell (H to 25.5 mm), whitish or variegated in color, sculptured by growth lines and sometimes presenting shallow varices. The shell is mainly characterized by the subovate peristome with an expanded and reflexed outer lip, a narrow and deep umbilicus and by the presence of a columellar fold.

   **Description:** Shell solid, of medium size (H up to 25.5 mm), elongated, fusiform; around 2.3 times as long as wide, rimate; upper whorls conic. Surface slightly shining; color white, corneous, or white with brownish axial streaks; sculptured by faint prosocline growth lines, crossed by minute and irregular spiral lines, forming a minutely reticulated surface in some areas. Irregular, longitudinal varices formed by old peristomes are occasionally found on the shell. Protoconch one and a half whorls, white to reddish-brown in color; smooth to the naked eye but actually sculptured with many small nodules and striations visible under microscope. Protoconch-teleoconch boundary well defined; teleoconch sculptured with fine growth lines and minor spiral lines better visible on earlier whorls; sculpture more distinct toward the umbilical area. Six and a half flat to slightly convex whorls; last

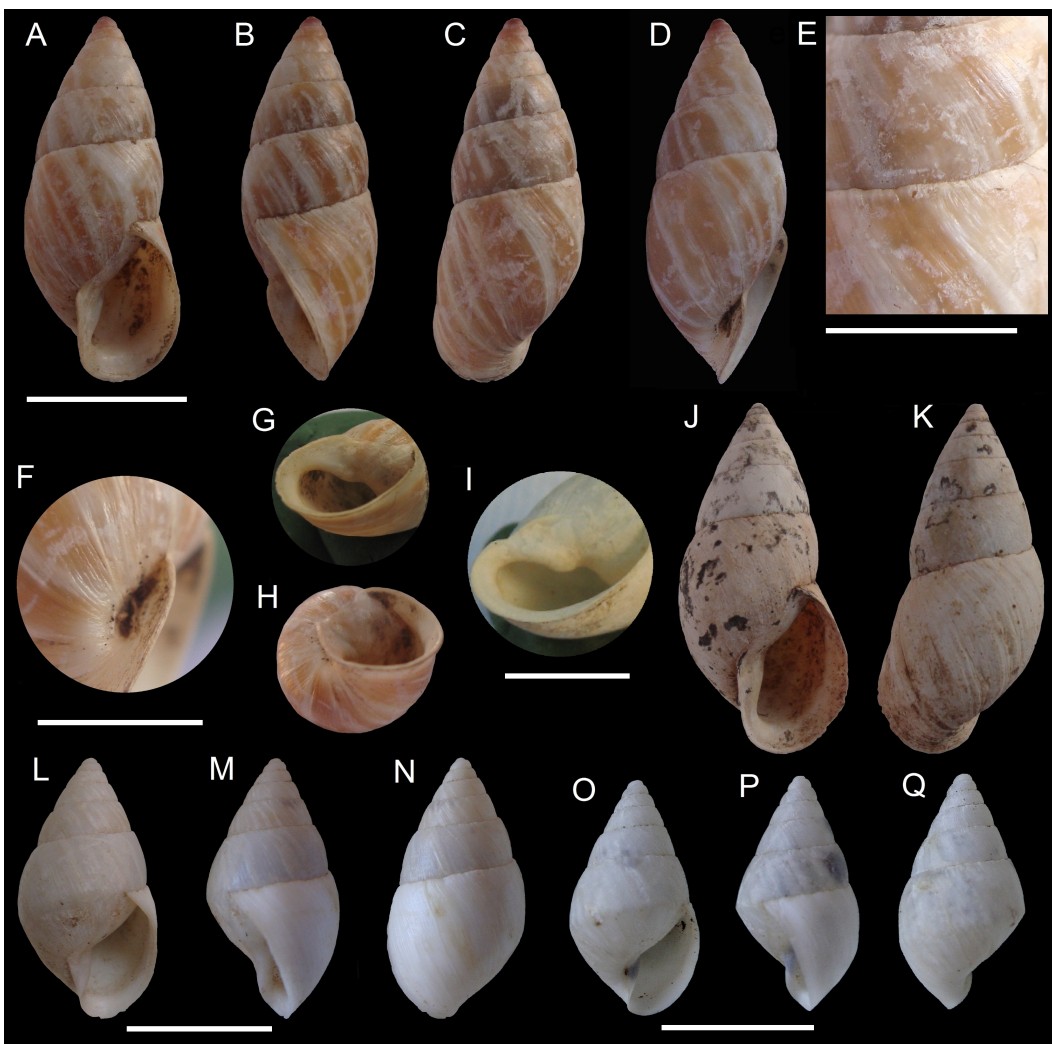

**Figure 4** *Scutalus chango* n. sp. Holotype MPCCL 020617, (A) apertural view, (B) side view (external lip view), (C) abapertural view, (D) side view (umbilical view), (E) detail of sculpture and sutures, (F) detail of umbilicus and columellar lip, (G) detail of columellar fold, (H) basal view; Paratype 5 MPCCL 030617D, (I) detail of columellar fold; Paratype 1 MPCCL 030617A, (J) apertural view, (K) abapertural view; Paratype 2 MPCCL 030617B (juvenile specimen), (L) apertural view, (M) side view (external lip view), (N) abapertural view; Paratype 3 MPCCL 030617C (juvenile specimen), (O) apertural view, (P) side view (external lip view), (Q) abapertural view. Scale bars are 10 mm for (A–D), (G–H), (J–K), (L–Q), and 5 mm for (E–F) and (I).

whorl convex and slightly angulated, about 0.66–0.68 of total height. Sutures impressed but shallow. Aperture large (HA about 0.44–0.48 of H), subovate (around 1.50–1.54 times as long as wide), slightly oblique and prosocline (about 27° with columellar axis). Columellar margin short, dilated above, minutely rugose, with a columellar fold in the interior of its upper part. Terminations of peristome joined by a moderately thin, oblique, parietal callus. Outer lip expanded and reflexed, sharp, often with the internal margin thickened. Umbilicus narrow and very deep. Soft parts unknown.

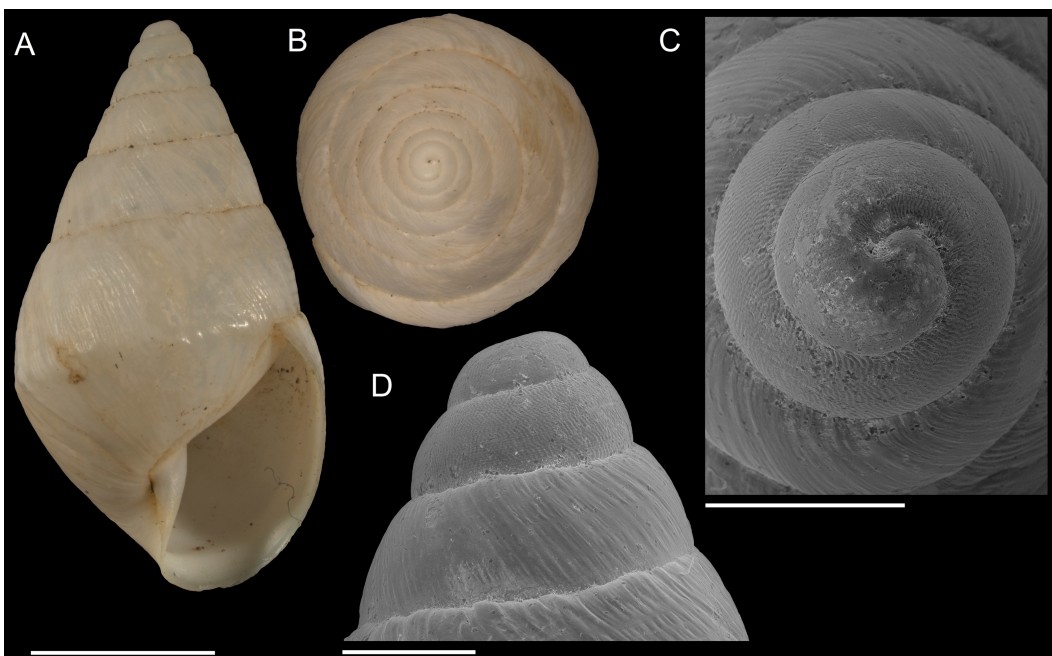

**Figure 5** *Scutalus chango* **n. sp. (A–D) paratype 4 SBMNH 460094, (A), apertural view, (B), apical view, (C), SEM side view of protoconch, (E), SEM apical view of protoconch.** Scale bars are 5 mm for (A) and (B), and 1 mm for (C) and (D).

**Type material:** Holotype MPCCL 020617 (Figs. 4A–4H): H: 24.8, HA: 10.6, LW: 16.6, NW: 7.5, SA: 47.5°, W: 10.7, WA: 7.3; paratype 1 MPCCL 030617A (Figs. 4J–4K): H: 25.3, HA: 12.4, LW: 16.8, NW: 7.5, SA: 50°, W: 11.2, WA: 8.1; paratype 2 (juvenile) MPCCL 030617B (Figs. 4L–4N): H: 18.1, HA: 9.4, LW: 12.9, NW: 7.0, SA: 59°, W: 9.2, WA: 6.2; paratype 3 (juvenile) MPCCL 030617C (Figs. 4O–4Q): H: 15.3, HA: 8.1, LW: 10.6, NW: 6.5, SA: 57°, W: 8.3, WA: 5.0; paratype 4 (juvenile) SBMNH 460094 (Figs. 5A–5D): H: 15.7, HA: 7.3, LW: 10.5, NW: 6.75, SA: 54°, W: 8.6, WA: 4.9; paratype 5 MPCCL 030617D (Fig. 4I): H: 25.4, HA: 12.6, LW: 17.2, NW: 7.5, SA: 47.67°, W: 11.2, WA: 8.1. All the specimens collected by M Araya and JF Araya, January 17, 2016.

**Type locality:** Foothills of the Chilean Coastal Range, north of Paposo (24°55′S; 70°30′W, altitude 150–170 m), Comuna de Taltal, Región de Antofagasta, northern Chile.

**Distribution and habitat:** Only known from type locality; shells found in humus under boulders and fallen rocks, usually near communities of the arborescent cacti *Eulychnia iquiquensis*, the large succulent shrub *Euphorbia lactiflua* and other small vegetation in the foothills of the Chilean Coastal Range. Many old shells and shell fragments were found buried in sediments in creeks and gullies, but no live specimens were recovered.

**Etymology:** A patronym (noun in apposition) in honor of the Chango people (now extinct) who inhabited coastal areas of northern Chile, having their last communities at Paposo, the type locality of the new species.

**Remarks:** Juvenile specimens have an obtusely angulated to almost carinated last whorl (Figs. 4L–4Q) and a rather narrow and slanted aperture (Figs. 4L and 5A), slightly semilunar

in some specimens (Fig. 4O); the external lip becomes reflexed and expanded, and the columellar lip widens in more mature specimens (Fig. 4L), while in fully mature shells the peristome is almost continuous, with a large, expanded and reflexed outer lip and a thin columellar fold (Figs. 4A and 4J). Evidence of episodic growth is seen in the irregular varices found in several specimens; this characteristic is unseen in any other Chilean terrestrial mollusk.

**Comparisons with related taxa:** This species differs from all other species of Chilean terrestrial snails by its slender shell, a flared and reflected apertural lip and by the presence of a columellar fold, a feature so far unique among Chilean terrestrial species. At first glance, this new taxon resembles *Scutalus latecolumellaris* Preston, 1909, which was reported by *Weyrauch (1967)* from northern Peru at an elevation of 1,700 m. However, the size difference (25 vs. 54 mm) immediately distinguishes both species. All other *Scutalus* species are decidedly stouter and cannot be confused with the new species. The protoconch of the type specimen of *S. latecolumellaris* (see Breure & Ablett, 2014: 106) differs slightly from this new species: "the sculpture [of *S. latecolumellaris*] looks more pitted than the wave like sculpture [in *S. chango*]" (J Ablett, pers. comm., 2017). Also, the protoconch of specimens of *S. proteus* Broderip, 1832 in the Leiden museum are slightly more pitted than in the new species. *Scutalus chango* sp. n. corresponds with the *Scutalus* species from northern-central Peru in the ecology and the occurrence in the low-altitude coastal area. We have also compared this new species to members of *Kuschelenia Hylton Scott, 1951*, found in high-altitude Andean areas in Argentina, Bolivia, Ecuador and Peru (*Hylton-Scott, 1951*; *Weyrauch, 1967*; *Breure, 1978*; *Breure, 1979*; *Miquel, 1998*; *Cuezzo, Miranda & Ovando, 2013*). The protoconch in *Kuschelenia* species has (anastomosing) axial wrinkles (*Breure, 1979*: 87), which are more prominent than in *S. chango* sp. n. The whorls of the new species are more convex, and the peristome more expanded than those in *Kuschelenia* species. The gap in distribution between *S. chango* sp. n. and other *Scutalus* species (northern Chile versus central and northern Peru; *Weyrauch, 1967*; *Breure, 1979*) is larger than the distribution gap between the new species and the nearest *Kuschelenia* species (northern Chile versus the altiplano of Bolivia). It cannot be excluded a priori that the new species may turn out to be a low-altitude representative of this genus. Although this remains somewhat puzzling, this can only be resolved in the future with anatomical and molecular data. For the time being, we prefer to consider the new species as a member of *Scutalus*. Finally, the shell shape of *S. chango* sp. n. is reminiscent of some *Drymaeus* species; however, the lattice sculpture of the protoconch, with axial riblets and spiral sculpture in the latter genus is clearly different from the protoconch sculpture in this new species. Type localities and records housed at the Leiden Museum of *Scutalus* and *Kuschelenia* species are shown in Fig. 1.

## DISCUSSION

The coastal areas of the Atacama Desert in northern Chile have been found to harbor a surprisingly rich diversity of land snails, almost matching the species richness of the much more humid Juan Fernandez Archipelago, off central Chile (*Miquel & Araya, 2013*;

*Araya & Catalán, 2014*; *Araya, 2015b*; *Miquel & Araya, 2015*). The areas near and around Paposo have previously yielded relatively rich snail harvests from early explorations, e.g., by Cuming (*Broderip & Sowerby, 1832a*; *Broderip & Sowerby, 1832b*) and the 'Comisión Científica del Pacífico' (*Hidalgo, 1869*); the latter collection recently revised by *Breure & Araujo (2017)*. In contrast with the much more arid inland areas of northern Chile, these coastal lowlands receive periodic fogs from the sea, which helps to sustain unique communities of plants in ravines and gullies in the West side of the Chilean Coastal Range. Taltal-Paposo in particular has a very rich diversity of endemic plant species, including some relict species with micro-ranges, acting as a local biodiversity island (*Ricardi, 1957*; *Dillon, 1991*; *Pizarro-Araya & Jerez, 2004*). The particular habitat of *S. chango* sp. n., living among and under large boulders, may provide microclimatic conditions similar to humid areas; this rock habitat is also relatively stable and buffered from climatic change. These litho-refugia have already been documented for Australia (*Couper & Hoskin, 2008*), and they may also explain the presence of charopid species in northern Chile, which require humid environments to thrive.

This fragile ecosystem is in peril due to urbanization and industrialization in the area, where a thermoelectric industry has already been established. Land snails are currently not taken into account in local governmental planning policies; a thorough evaluation proper knowledge of the species present in northern Chile and of their distributions is essential for future conservation efforts, especially in hotspots of biodiversity like Paposo.

## CONCLUSIONS

A new terrestrial bulimulid species (Gastropoda: Orthalicoidea), *Scutalus chango* sp. n., is described from Paposo, Región de Antofagasta, northern Chile, being the first record of the genus *Scutalus* in Chile and the southernmost record for this endemic South American genus. The new species may represent part of a relict fauna at the coastal area of northern Chile, with close relationship with species from central-northern Peru.

### Abbreviations

| | |
|---|---|
| H | height (maximum dimension parallel to axis of coiling) |
| HA | height of aperture |
| HS | height of spire; |
| LW | height of last whorl |
| SA | spire angle |
| W | width (maximum dimension perpendicular to H) |
| WA | width of aperture |

## ACKNOWLEDGEMENTS

We are thankful to Daniel Geiger and Vanessa Delnavaz (SBMNH, Santa Barbara, USA) for their help with the SEM and light microscopy images of the specimens, to Jonathan Ablett (Natural History Museum, London, UK) for kindly reviewing the English of the final manuscript, and to Marta Araya (Caldera, Chile) for her help in the field work in

Paposo. We thankfully acknowledge the constructive comments of the reviewers, Frank Köhler (Australian Museum, Sydney, Australia) and Igor Muratov (Muséum National d'Histoire Naturelle, Paris, France), and of the editor Rudiger Bieler (Field Museum of Natural History, Chicago, USA).

### Funding
The authors received no funding for this work.

### Competing Interests
The author declares that they have no competing interests.

### Author Contributions
- Juan Francisco Araya conceived and designed the experiments, performed the experiments, analyzed the data, contributed reagents/materials/analysis tools, wrote the paper, prepared figures and/or tables, reviewed drafts of the paper.
- Abraham S.H. Breure performed the experiments, analyzed the data, contributed reagents/materials/analysis tools, wrote the paper, prepared figures and/or tables, reviewed drafts of the paper.

### Data Availability
The specimens for study are deposited in the collections of the Museo Paleontológico de Caldera (MPCCL), in Caldera, Chile and in the Santa Barbara Museum of Natural History (SBMNH) in Santa Barbara, USA.

### New Species Registration
The following information was supplied regarding the registration of a newly described species:

*Scutalus chango* sp. n.: urn:lsid:zoobank.org:act:E70CF791-7574-45C4-A393-B5F4F3C6E94B.

Publication LSID: urn:lsid:zoobank.org:pub:C9BE441E-6159-4973-888D-74660B2C25F3.

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
