# Peer review of "A new terrestrial snail species (Gastropoda: Bulimulidae) from the Región de Antofagasta, northern Chile"

_PeerJ, doi:10.7717/peerj.3538_

## Round 0.1 · original submission · Minor Revisions

Your manuscript has been read by two reviewers who recommend some – mostly minor – additions and modifications. I agree with their assessments. Please provide a more detailed comparison and discussion of morphological characters (be more specific than merely stating that the protoconch is “slightly different” or that the shells are “stouter”). Add a more extensive discussion why this could not be a member of Kuscheliana (and perhaps, as suggested by one of the reviewers, compare it also to Drymaeus). Consider adding a comparative table of characters if that would help with making the case. Add an illustration of the columellar fold if that is indeed an important distinguishing character. Also, I suggest making it clearer up front (already in the Abstract) that this study is based on empty shells alone. When I started reading the manuscript, I was wondering about molecular approaches -- until I learned (in the Material & Methods section) that this would not be possible.

I noticed a few oddities in the text (such as “knowledge on its diversity”) and the reviewers pointed out a few others. Please have the manuscript read by a native English speaker. Unfortunately, PeerJ cannot offer copyediting services.

·

Basic reporting

The manuscript contains the description of a new species of land snails from northern Chile.

Use of English language is sufficient; I suggested some corrections where considered useful or necessary.
Please refrain from using terms, such as "unusual" or "conspicuous" lightly, unless the species is indeed unusual. I don't see what makes the present species 'unusual'. If it is, please explain.

Relevant literature is cited.

Experimental design

The methods used are basic but appropriate; described with sufficient detail.

In your taxonomic part, under genus please provide a generic diagnosis, especially with regard to how differentiate Scutalus from Kuschelenia. You need to provide sufficient evidence for placement in the former and not the latter given the large disjunction in the distribution of Scutalus that your work proposes.

At present you only allude to the fact that Scutalus species are found at low elevations, while Kuschelenia species are found at high elevations. That makes me wonder, why cant the new species just be a lowland form of Kuschelenia?

Validity of the findings

The novelt of the species seems plausible given the evidence put forward.
However, the comparison with already known species of this genus could have been a little more transparent and 'objective'. Instead of saying 'all other species of the genus are stouter and cannot be mistaken with the new species'.
Its nice that you are confident, but it would be more scientific if you quantify the differences or at least document them instead of leaving the reader in a situation of "trust me, I know it". One could have used measurements to qualify how much stouter they are or photographs to document the differences, etc. at least literature references to descriptions of other species.

May I comment that one needs to be careful with using shell characters in species that inhabit extreme habitats. There are ample of examples from other land snails that under extreme conditions shell characters may vary even within a species, particularly shell shape and size, as they are under strong local selection. A good example is the study of Rhagada species from Rosemary Island (Australia), where extreme shell forms are likely expressed by the same species along a very steep ecological gradient on very narrow geographical scales. Hence, just that a snail differs in shell shape or size is a very poor indicator for its distinctiveness of already names species. I am only convinced that this is likely a new species because it appears to be very isolated from its congeners.

Additional comments

Avoid arm waiving. Be concise and use data instead of personal experience to underpin the results of your work.

Please refer to the annotations in the text for further comments. I wonder whether you should name the new species 'changos' (for the people), not just 'chango' (for one person?)

·

Basic reporting

Some corrections in English grammar and style are needed. Please see attached PDF.

Experimental design

Coordinates for the type locality are highly inaccurate. More than 1300 meters error. Correct coordinates are: 24° 55.74'S; 70° 30.05'W.

Validity of the findings

No comment.

Additional comments

Would it be possible to compare proposed new species with Drymaeus in this publication?

---

## Round 0.2 · accepted · Accept

Thank you for your revised manuscript. I am satisfied with your response to the referee comments, and am now happy to move this forward into production.

---

## Author Rebuttal · Round 0.2

Concepción, June 12, 2017

Dear Colleagues:

Thank you for the time and effort spent in editing and correcting our manuscript. I am here presenting the revised text: we have acknowledged most, if not all, of the suggestions and corrections and in the next lines we have answered point by point all of those. I have uploaded a revised version of the manuscript (with track changes) incorporating all the changes provided in both PDFs by the reviewers, and we am presenting details on each correction below. Please, let me know if there are any other queries regarding this submission. Thanks again for your collaboration.

Best regards,

Juan Francisco.

**Comments from Editor**

Your manuscript has been read by two reviewers who recommend some – mostly minor – additions and modifications. I agree with their assessments. Please provide a more detailed comparison and discussion of morphological characters (be more specific than merely stating that the protoconch is "slightly different" or that the shells are "stouter"). Add a more extensive discussion why this could not be a member of Kuscheliana (and perhaps, as suggested by one of the reviewers, compare it also to Drymaeus). Consider adding a comparative table of characters if that would help with making the case. Add an illustration of the columellar fold if that is indeed an important distinguishing character. Also, I suggest making it clearer up front (already in the Abstract) that this study is based on empty shells alone. When I started reading the manuscript, I was wondering about molecular approaches -- until I learned (in the Material & Methods section) that this would not be possible.

**Response: We have modified the discussion of the manuscript incorporating more details on the comparison with related taxa, also including *Drymaeus* and *Kuschelenia*. We have also added additional images of the columellar fold and stated clearly that the description is based on empty shells.**

I noticed a few oddities in the text (such as "knowledge on its diversity") and the reviewers pointed out a few others. Please have the manuscript read by a native English speaker. Unfortunately, PeerJ cannot offer copyediting services.

**Response: We have checked the manuscript again, and it has been revised by a native English speaker (Dr. Jonathan Ablett, Natural History Museum, London, UK)**

**Comments from Reviewers**

**Reviewer: Frank Koehler**

Basic reporting
The manuscript contains the description of a new species of land snails from northern Chile.

Use of English language is sufficient; I suggested some corrections where considered useful or necessary.
Please refrain from using terms, such as "unusual" or "conspicuous" lightly, unless the species is indeed unusual. I don't see what makes the present species 'unusual'. If it is, please explain.

Relevant literature is cited.

Experimental design
The methods used are basic but appropriate; described with sufficient detail.

In your taxonomic part, under genus please provide a generic diagnosis, especially with regard to how differentiate Scutalus from Kuschelenia. You need to provide sufficient evidence for placement in the former and not the latter given the large disjunction in the distribution of Scutalus that your work proposes.

At present you only allude to the fact that Scutalus species are found at low elevations, while Kuschelenia species are found at high elevations. That makes me wonder, why cant the new species just be a lowland form of Kuschelenia?
**Response: We have added a generic diagnosis for *Scutalus* and a further discussion on the generic allocation of the new species.**

Validity of the findings
The novelt of the species seems plausible given the evidence put forward.
However, the comparison with already known species of this genus could have been a little more transparent and 'objective'. Instead of saying 'all other species of the genus are stouter and cannot be mistaken with the new species'.
Its nice that you are confident, but it would be more scientific if you quantify the differences or at least document them instead of leaving the reader in a situation of "trust me, I know it". One could have used measurements to qualify how much stouter they are or photographs to document the differences, etc. at least literature references to descriptions of other species.

May I comment that one needs to be careful with using shell characters in species that inhabit extreme habitats. There are ample of examples from other land snails that under extreme conditions shell characters may vary even within a species, particularly shell shape and size, as they are under strong local selection. A good example is the study of Rhagada species from Rosemary Island (Australia), where extreme shell forms are likely expressed by the same species along a very steep ecological gradient on very narrow geographical scales. Hence, just that a snail differs in shell shape or size is a very poor indicator for its distinctiveness of already names species. I am only convinced that this is likely a new species because it appears to be very isolated from its congeners.
**Response: We have provided more details on the comparison of the new species with similar *Scutalus* in the area.**

Comments for the author
Avoid arm waiving. Be concise and use data instead of personal experience to underpin the results of your work.

Please refer to the annotations in the text for further comments. I wonder whether you should name the new species 'changos' (for the people), not just 'chango' (for one person?)

**Comments on the annotated manuscript (PDF):**

Commented [FK1]: I am personally not a fan of using such attributes in a scientific text unless they are really justified. I am wondering, why is this species conspicuous? Or is it just new?
**Response: we have decided to delete this word.**

Commented [FK2]: Again. What makes this species so unusual? Please explain!
**Response: we have decided to delete this word.**

Commented [FK3]: This is a very long sentence.. break up into two
**Response: we have re-worded this sentence.**

Commented [FK4]: Provide complete reference
**Response: we have now provided complete references.**

Commented [FK5]: What does this mean? I don't know this word.
**Response: 'rimate' is not unusual in the group (Pilsbry already did use it), meaning 'very narrowly umbilicated'.**

Commented [FK6]: Shouldn't it be 'changos' then? For the people (not just one person)
**Response: we have changed this word.**

Commented [FK7]: I don't like this word. Perhaps just use the name of the new species or 'new species'.
**Response: we have modified this word in the text.**

Commented [FK8]: Be precise and say exactly how it differs. Its not good enough just tu assure the reader that it differs without providing the data that underpins this claim
**Response: we have modified this paragraph extensively to address this issue.**

Commented [FK9]: 'point more' is a rather vague statement! So you are not confident about the genus placement? Please be honest and explain why this is the case. After disclosing the ambiguity, you may make an informed guess, but call it out so it's clear.
**Response: same as above.**

Commented [FK10]: That's always the case
**Response: we deleted this text.**

Commented [FK11]: Make sure the reference complies with the reference guide of the journal. Martens is the editor of this work.
**Response: we have corrected the references.**

### Reviewer: Igor Muratov

Basic reporting
Some corrections in English grammar and style are needed. Please see attached PDF.

Experimental design
Coordinates for the type locality are highly inaccurate. More than 1300 meters error.
Correct coordinates are: 24° 55.74'S; 70° 30.05'W.
**Response: We did this to avoid the possibility of the locality being raided by (commercial) shell collectors, see below.**

Validity of the findings
No comment.

Comments for the author
Would it be possible to compare proposed new species with Drymaeus in this publication?
**Response: We have added some remarks on the discussion regarding Drymaeus and the new species.**

**Comments on the annotated manuscript (PDF):**

Type species of *Scutalus* is from Baja California (North America).
**Response: Apparently, the type species of *Scutalus* (*Scutalus proteus*) is native to Peru. *Scutalus* is so far restricted to western South America (See Breure, 1978; Breure & Ablett, 2014).**

"are studies" or just "are"
**Response: done.**

, which
**Response: done.**

as well as
**Response: done.**

narrow distribution
**Response: done.**

"species" cannot be "with …high levels of endemism"
**Response: we have changed the sentence.**

… fold. It was collected …
**Response: done.**

Humus: "organic component of soil, formed by decomposition of plant and animal material".
Soil: "A mixture of organic remains, clay, and rock particles".
**Response: it was indeed humus.**

as well as
**Response: we have re-worded this sentence.**

Type species of *Scutalus* is from Baja California (North America).
**Response: see above**

of
**Response: done.**

that
**Response: done.**

Humus: "organic component of soil, formed by decomposition of plant and animal material".
Soil: "A mixture of organic remains, clay, and rock particles".
**Response: it was indeed humus.**

Highly inaccurate. More than 1300 meters error. Correct coordinates are: 24° 55.**74**'S; 70° 30.**05**'W.
**Response: We did this expressly so, to avoid the possibility of the locality being raided by (commercial) shell collectors. We have added "(precise locality data available from first author on request)" in the text.**

and include
**Response: done.**

More than one?
**Response: done (singular: name).**

mainly
**Response: done.**

If this character is so important and "so far unique among Chilean terrestrial species" (as it is stated below), then why it is not illustrated? Columellar margin looks just slightly arcuate on the photos, without any fold.
**Response: done, we have added additional figures depicting this character.**

actually sculptured with many small nodules and striations visible under microscope.
**Response: done.**

better
**Response: done.**

on
**Response: done.**

"distinct"?
**Response: done.**

Inconsistent: "HA" on figure 3.
**Response: done, we have checked consistency in the text.**

Of H
**Response: done.**

Concave? In which part?
**Response:  we have modified this text.**

Not visible on any presented illustrations.
**Response: We have added an additional illustration of this character.**

part
**Response: done.**

Inconsistent: "telegraph" style elsewhere. Remove "the" and "an".
**Response: done.**

Highly inaccurate. More than 1300 meters error. Correct coordinates are: 24° 55.**74**'S; 70° 30.**05**'W.
**Response: see above.**

Humus: "organic component of soil, formed by decomposition of plant and animal material".
Soil: "A mixture of organic remains, clay, and rock particles".
**Response: it was indeed humus.**

How many? Are any of them not paratypes? If so, then they should be listed under "other material examined".
**Response: We have added the required text.**

inhabited coastal
**Response: done.**

**Juan Francisco Araya & Abraham Breure**